# Heart Donation and Preservation: Historical Perspectives, Current Technologies, and Future Directions

**DOI:** 10.3390/jcm11195762

**Published:** 2022-09-28

**Authors:** Nicholas R. Hess, Luke A. Ziegler, David J. Kaczorowski

**Affiliations:** 1Division of Cardiac Surgery, Department of Cardiothoracic Surgery, University of Pittsburgh School of Medicine, Pittsburgh, PA 15213, USA; 2University of Pittsburgh Medical Center Heart & Vascular Institute, Pittsburgh, PA 15215, USA

**Keywords:** heart transplantation, heart preservation, heart perfusion, heart donation, technology

## Abstract

Heart transplantation has become the accepted treatment for advanced heart failure, with over 4000–5000 performed in the world annually. Although the number of yearly transplants performed has been increasing over the last decade, the number of candidates in need of transplantation continues to grow at an even faster rate. To distribute these scarce and precious resources equitably, donor heart placement is based on clinical need with priority given to those who are more critically ill. As a result, donors are matched with recipient candidates over increasingly farther distances, which may subject these organs to longer ischemic times. One of the mainstays of successful heart transplantation is successful organ preservation while the donor organ is ex vivo from the time of donor procurement to recipient implantation. In order to adapt to a new era of heart transplantation where organs are shared across wider ranges, preservation strategies must evolve to accommodate longer ischemia times while mitigating the harmful sequalae of ischemia-reperfusion injury. Additionally, in order to address the ever-growing supply demand mismatch of donor organs, evolving perfusion technologies may allow for further evaluation of donor grafts outside of conventional acceptance practices, thus enlarging the effective donor pool. Herein this review, we discuss the history of organ preservation, current strategies and modalities employed in current practice, along with developing technologies in preclinical stages. Lastly, we introduce the concept of donation after circulatory death (DCD), which has been until recently a largely unexplored avenue of heart donation that relies much on current preservation techniques.

## 1. Introduction

Heart transplantation is the gold standard treatment for end-stage heart failure [1]. Each year, the annual number of heart transplants has grown to over 5000 world-wide [2], with over half performed in the United States [3]. Currently, the number of annual transplants performed is limited by the availability of donor organs, as the demand greatly exceeds supply. Donor-recipient pairing is limited in large by geographical constraints, even more so than other organs such as kidney and liver, as donor hearts are not able to tolerate long periods of ischemia induced by organ transport [4,5]. Increasing ischemic times have been associated with higher risk of early graft failure, impaired graft function [6], and posttransplant mortality [7,8]. As a general rule, ischemia times for donor hearts should be kept below 4–6 h to minimize these risks. Because of these rigorous time constraints, there is evidence that geographic disparities continue to exist with respect to organ access, even despite recent allocation policy changes aimed toward addressing them [9]. These geographical constraints are largely driven by travel distances, and ultimately ischemia times needed to transport a donor organ to the recipient hospital. Thus, this narrow range of acceptable ischemia time remains one of the largest limitations for optimal donor-recipient pairing worldwide.

In addition to careful donor/recipient selection, operative technique, and posttransplant care, a crucial yet often overlooked component of transplantation is successful preservation of the donor organ while it is transported to the recipient’s transplanting facility. Organ preservation aims to provide a means of effective organ transport while reducing the harmful sequalae of organ ischemia and hypoxia while the organ is ex vivo, as well as subsequent reperfusion injury. In light of recent allocation policy changes, organ transport distances and ischemia times have increased [10]. Additionally, the use of organs from increasingly older donors has been observed worldwide, especially among European countries [11], which have been shown to be more sensitive to longer ischemia times than hearts obtained from younger donors [12,13]. Given these findings, much recent clinical interest has been focused on improved organ preservation and ex vivo heart perfusion techniques as means of minimizing the impacts of graft ischemic injury, and safely extending acceptable ischemia times [14,15]. In this review, we will describe the tenets of heart preservation, as well as reflect upon the history of its development and refinement since the beginning of the transplantation era. Furthermore, we will describe the most common modalities utilized in current clinical practice. Lastly, we will introduce the most recent advancements in preservation technology aimed at extending safe preservation times and evaluation of donor hearts outside the standard criteria of transplant. The goals of these technologies are to not only reduce the harmful, lasting effects of ischemia, but to also reduce the existing geographic disparities and expand the current donor pool by perhaps enabling use of hearts from more marginal and older donors that would otherwise not be felt to tolerate ischemia times necessary for transportation and delivery to the recipient hospital.

## 2. Historical Aspects of Cardiac Donation

In 1967, the first human-to-human heart transplant was performed by Dr. Christiaan N. Barnard in Cape Town, South Africa [16]. The donor for this operation, a young girl struck by a car, was matched to a 54-year-old male with ischemic cardiomyopathy. The pronouncement of death was made after the donor’s ventilator was disconnected, and once electrocardiographic activity had ceased for 5 min. Following this, the heart was re-perfused by placing the donor on cardiopulmonary bypass and subsequently harvested. After donor cardioectomy, the donor heart was perfused via an aortic cannula with flow into the aortic root and coronary arteries until the recipient cardioectomy was completed and the donor heart could be implanted. To limit transport and heart ischemia times, the donor and recipient were in adjacent operating theatres. The following early series of heart transplants were performed in a similar fashion using donation after cardiac death (DCD); however, survival rates were poor [17].

In 1968, Henry Beecher and the Harvard Ad Hoc Committee began to lay the framework surrounding the legal definitions of death, most importantly by suggesting that “irreversible coma” be included as a criterion for death [18]. By 1981, the concept of brain death had been written into every state’s collective legislature with the adoption of the Uniform Determination of Death Act [19]. Since then, donation after brain death (DBD) has become the mainstay method of heart procurement and transplantation. Advantages of this method of donation include the ability to achieve a quick cardioplegic arrest in a controlled fashion without subjugation of the heart and other organs to prolonged warm ischemia. Additionally, diastolic arrest via cold cardioplegia and storage of the heart in a hypothermic state (discussed later) would then allow for transport between hospital facilities. In 1979, the Stanford group suggested that donor hearts could be procured outside of the recipient hospitals, quoting equivalent outcomes, albeit longer ischemia times, among 19 distally harvested hearts in comparison to 20 harvested within the recipient hospital [20]. These findings have allowed for a larger network of organ sharing and expansion of the effective donor pool.

The concepts of DBD donation, cold storage and inter-facility transport of donor hearts, along with the implementation of cyclosporine into heart transplantation in 1980 [21] led to a revitalization of the field and a rapid increase in the number of annual cases performed. After the adoption of DBD donation for heart transplantation, use of DCD donors for heart transplantation was largely abandoned. However, in recent years, clinical interest in DCD donation has been reignited [22] (described later in this article) as a response to an increasing heart failure population and a largely stable, yet inadequate number of suitable donor organs. In this article, we will describe the current methods of heart preservation as they apply to DBD method of donation. We will then discuss the current implications of DCD donation in heart transplantation and specific aspects of organ preservation that apply to this mode of organ procurement.

## 3. Current Methods of Heart Preservation

### 3.1. Static Cold Storage

Static cold storage is the most commonly utilized method of heart preservation, both in the United States and the world and for both adult and pediatric hearts. This method is predicated on rapid removal of blood from the organ, complete washout of the vascular bed with a heart preservation solution, and maintenance of the organ in a state of hypothermia until it can be transported to the recipient hospital and implanted. Despite small differences in operative technique, most centers and transplanting surgeons follow a similar sequence for donor cardioectomy.

After the donor heart is inspected and accepted for transplant, the cardioectomy operation commences, usually in concert with other organ procuring teams. The superior vena cava is ligated, and the inferior vena cava is incised to exsanguinate the patient. Following this, the left heart is vented, most commonly via either incision of the left atrial appendage or a pulmonary vein. The aorta is then cross-clamped, and the heart is flushed with cold heart preservation solution delivered through an aortic root cannula and cooled with topical ice. Once the heart has been cooled and flushed, it is explanted from the body [23]. The heart is then typically placed into a sterile bag filled with preservation solution, and then bagged twice more and transported in an ice-filled cooler. These measures aim to achieve a diastolic arrest of the heart, decrease its metabolic demands, and minimize the harmful effects of ischemia while in transport.

Hypothermia does not halt cellular metabolism entirely. However, it does slow the rate at which most human enzymes degrade compounds necessary for cell viability [24]. Furthermore, hypothermia reduces the rate of lysis of intracellular organelles such as lysosomes, preventing release of autolytic enzymes and resultant cell death. According to the Vant Hoff’s rule, a temperature-dependent reduction metabolic activity occurs with cooling, resulting in a 50% reduction in metabolic activity for every 10 °C reduction in temperature [25]. At 4 °C, metabolic activity is approximately 10–12% of baseline normothermic conditions [26].

Unwanted side effects of prolonged ischemia and hypothermia include cellular edema [27], acidosis [28], and generation of reactive oxygen species upon re-perfusion [29]. In order to mitigate these effects, numerous heart preservation solutions have been developed, each with differing concentrations of cellular nutrients and metabolites, electrolytes, buffering agents, and antioxidants. Euro Collins solution, developed in the 1960s, was the first and only available preservation solution for approximately 15 years until the University of Wisconsin (UW) solution was introduced in 1988 [30]. Development of new and alterations of existing solutions increased exponentially following this. In fact, during the 1990′s, over 150 different solutions were utilized clinically among 147 United States transplanting centers. [31,32] In today’s practice, the three most commonly employed heart preservation solutions include histidine-tryptophan-ketoglutarate (HTK) solution, University of Wisconsin (UW) solution, and Celsior solution. Compositions of these solutions are presented in Table 1 [33,34,35].

Multiple studies have compared posttransplant outcomes with various preservation solutions. However, the results are conflicted [36]. A study of 64 hearts preserved with Celsior were found to have no differences in 30-day mortality when compared a control group of 67 hearts preserved with any other available solutions [37]. Another study found no differences in in-hospital mortality, or graft failure among hearts preserved with Celsior (*n* = 38), HTK (*n* = 61), or Plegisol (*n* = 34) [38]. In a series by Kofler et al., [39] one-year posttransplant survival improved after switching from HTK to UW solution; however, this study may have been influenced by certain temporal biases. Another small comparative study comparing Celsior and UW found no statistically significant differences in one-year survival or primary graft dysfunction [40]. In the largest comparative series, UW (*n* = 3107) was shown to have a statistically significant higher rate of one-year survival over Celsior (*n* = 1803), though the differences were clinically small (89.6% vs. 87.0%, *p* < 0.01) [41].

Regardless of commercially available heart preservation solution used, the results of static cold storage are optimal when ischemic times are less than 4–6 h. After this timepoint, risk of posttransplant primary graft failure and death increases significantly [12,32,42,43,44]. Though some centers have reported acceptable outcomes with ischemic times >6 h [45,46], these results may only be applicable to very carefully selected donor-recipient pairs.

### 3.2. Paragonyx SherpaPak

Paragonyx (Paragonyx Technologies Inc., Braintree, MA, USA) has developed the SherpaPak system. This device is based on the premise that the ideal storage temperature of organs resides between 4 °C and 8 °C. This temperature range is derived from studies that demonstrate that storage of organ tissue within this range may decrease metabolic demand and hypoxic injury related to ischemia, while also preventing cellular damage and protein denaturization associated with colder temperatures [47,48,49,50,51]. This technology differs from the conventional practice of cold ice storage, where donor grafts are often kept at a temperature of below this optimal window [52]. With conventional cold storage, the temperature of the organ may be heterogeneous and freezing of tissues with resultant cellular injury may occur.

The Paragonyx SherpaPak Cardiac Transport System is a single-use, disposable and non-perfusing storage system designed to maintain donor heart temperatures between 4 °C and 8 °C for extended periods of time (Figure 1). In this system, the donor heart is harvested, affixed to the device’s connecter and temperature probe, and placed into the inner cannister. The inner cannister is then filled with cold cardioplegia preservation solution and de-aired, allowing full submersion of the heart within the preservation solution. In preclinical and clinical studies, Celsior, UW, and HTK solutions have all been utilized with this system [53,54,55]. The inner cannister is then placed within the outer cannister, surrounded by disposable cooling packs. Unlike normal ice which undergoes phase change at 0 °C, these cooling packs undergo phase change at 5 °C, maintaining goal preservation temperatures. Preclinical work has demonstrated the SherpaPak can maintain optimal temperature ranges for 30 h, even in varying outer environmental temperatures [54]. Furthermore, pig grafts were able to be cooled within desired temperature ranges for 12 h [43]. The first clinical application of this system was performed by Naito et al. who used the SherpaPak to transport a donor organ 1100 miles to the recipient institution with cold ischemia time of 312 min [53]. In an early clinical trial, 7 donor human hearts were successfully preserved and transplanted with the SherpaPak system. When compared to 14 matched recipients who received hearts that were preserved with conventional ice storage, rates of early graft failure were comparable [55]. Thirty-day and one-year outcomes were also similar, with thirty-day mean right ventricular function modestly better in the SherpaPak group (tricuspid annular plane systolic excursion 17.83 vs. 14.42 mm, *p* = 0.02). One-year mean left ventricular ejection fraction was modestly improved in the SherpaPak group (62.40% vs. 55.27%, *p* = 0.03). Use of this system clinically in pediatric heart transplant is also underway [56].

### 3.3. Transmedics Organ Care System

Early experience with normothermic, beating-heart perfusion for preservation of donor hearts during distant procurement was described in the 1980s by Hardesty and Griffith in the context of combined heart and lung transplant. After initial laboratory success with canine and bovine heart-lung blocs [56] they employed this technique clinically, reporting use in 20 patients [57].

In current practice, the first and only current commercially available platform that offers normothermic, beating-heart perfusion is the Transmedics Organ Care System (OCS) (Transmedics Inc., Andover, MA, USA) (Figure 2). This platform has been recently approved by the United States Food and Drug Administration following clinical trials. Clinical interest with this device is multi-faceted, centering upon expansion of the preservation window, evaluation of extended-criteria donors, and re-perfusion of DCD donor hearts.

#### 3.3.1. Setup of the OCS [58,59]

During donor heart procurement, the donor heart is dissected with isolation of the main vessels. The OCS system is then primed with a nutrient-rich solution. After donor heparinization, donor blood is extracted from the patient in order to serve as perfusate for the OCS platform. This blood can be extracted from the patient via a cannula temporarily inserted into the right atrium. Following blood donation into the OCS, the cannula may be removed and the harvesting of the donor heart may proceed under standard protocol. The aorta is cross-clamped and both topical ice and antegrade cardioplegia delivered via the aortic root are applied to induce diastolic arrest. Following cardioplegia, the cardioectomy is performed in usual bicaval fashion.

Following cardiectomy, the donor graft is prepared under cold ischemia prior to connection and reperfusion with the OCS system. First, an aortic cannula is secured to the donor aorta. This cannula will perfuse the aortic root, and thus, the coronary arteries with nutrient-enriched donor blood. Second, an additional cannula is inserted into the right ventricle via the pulmonary artery and secured into place. This cannula collects coronary sinus return, and allows for samples to be drawn for blood lactate measurements. Lastly, a third cannula may be inserted into the left atrium for left ventricular venting. The cannulas are then connected to the OCS system, and the heart is perfused via the aortic cannula. Once the donor graft is perfusing and reaches normothermia, the goal is to reestablish sinus rhythm. This may occur spontaneously, or require either cardioversion or placement of epicardial pacing wires. With the heart beating, the inferior and superior vena cava may be closed with either sutures or a silk tie. The OCS system can then be sealed, and the perfusion parameters adjusted via the system’s console. Additionally, a side port allows for blood gas and lactate samples to be drawn and trended during transport. Of measurable parameters, serum lactate has been shown to be the most sensitive and specific predictor of early graft failure [60].

#### 3.3.2. Clinical Usage and Applications

During its early clinical applications, the OCS was first evaluated among cardiac grafts from standard donors. In a prospective, non-randomized trial performed between 2006–2008, 29 patients underwent cardiac transplantation after donor graft preservation with the OCS [61]. Compared to 130 control patients using static cold storage preservation, 30-day, 1-year, and 2-year survival were similar. Additionally, hospital length of stay was comparable between cohorts. Though not significant, there was a trend of decreased primary graft failure (6.9% vs. 15.3; *p* = 0.20) and severe acute rejection (17.2% vs. 23.0%; *p* = 0.73) in those transplanted with grafts preserved with OCS. The PROCEED II trial (2010–2013) was a randomized, noninferiority trial conducted that compared 67 OCS-perfused grafts with 63 cold storage grafts [62]. Marginal donors were excluded from this trial. The primary endpoint of 30-day mortality did not significantly differ between groups (OCS 94%; cold preservation 97%). Likewise, the secondary endpoints of intensive care length of stay, incidence of severe rejection episodes, and cardiac-related serious adverse events were similar between cohorts. Between cohorts, the OCS was associated with longer overall preservation times (324 min versus 195 min, *p* < 0.001), however, cold ischemia times were shorter (113 min versus 195 min, *p* < 0.001). One center that performed 38 transplants (19 OCS, 19 cold storage) as part of the PROCEED II trial has published 2-year outcomes [63]. In this analysis, OCS was associated with a lower, but not significant difference in 2-year survival in comparison to static cold storage (72.2% versus 81.6, respectively, *p* = 0.38). Secondary outcomes at 2-years including freedom from chronic allograft vasculopathy, non-fatal major cardiac adverse events, biopsy-proven cellular rejection, and antibody-mediated rejection were similar between cohorts. Use of the OCS system has also been described for standard pediatric cardiac grafts in patients requiring complex concomitant procedures with the risk of prolonging ischemic times. Fleck and colleagues reviewed 8 children transplanted with grafts persevered using OCS compared with 13 children transplanted using static cold storage [64]. The incidence of primary graft failure, as well as the graft function and occurrence of any treated rejection at follow-up revealed no significant difference between the two groups.

In addition to preservation of standard donor grafts, clinical interest has been focused on utilization of the OCS for marginal donors or those with anticipated long preservation times. Such efforts are aimed towards expanding the current restrictive and limiting donor pool. The EXPAND Trial was a multi-center study conducted to evaluate the efficacy of the OCS in assessing, preserving, and possibly resuscitating hearts outside the normal acceptance criteria [65]. Ninety-three hearts were enrolled with either expected total ischemic time ≥4 h, or >2 h plus at least one of the following: (1) left ventricular hypertrophy, (2) ejection fraction 40–50%, (3) circulatory downtime ≥20 min, or (4) age >55 years. A total of 81% of these hearts were utilized for transplant, and 30-day and 6-month posttransplant survival were 94.7% and 88.0%, respectively. 24-h rate of severe primary graft dysfunction was 10.7%.

Individual institutions have also reported experiences with utilization of the OCS for extended-criteria donors. Garcia Sáez and colleauges [58] reported a series of 30 marginal donor hearts transplanted after preservation with OCS and mean total preservation time of 284 min. At a mean follow-up of 257 days, overall posttransplant survival was 96.2%. Over this period, 4 hearts were declined after perfusion with OCS due to poor contractility and/or rising serum lactate levels. This group also reported another series of 30 patients who underwent bridge to transplantation with a left ventricular assist device (LVAD) via sternotomy Fifteen of these patients received a graft preserved with the OCS, and static cold storage preservation was used in the other 15 [66]. Donors in this study met standard acceptance criteria, but with anticipated prolonged preservation time due to recipient resternotomy and LVAD explantation. Total out-of-body time was longer with OCS preservation (312 min vs. 204 min, *p* = 0.021), but with shorter cold ischemic time (89 min versus 204 min, *p* < 0.001). Those with OCS preservation experienced improved 30-day survival (100% versus 73.35%, *p* = 0.03). Wong and colleagues [67] reported a series of 9 deployments of OCS during a 2-year period (2017–2019). OCS was utilized when the donor displayed any of the following four features: (1) anticipated cold ischemic time 4–8 h, (2) age >55 but with normal coronary angiography, (3) circulatory down-time >20 min, or (4) left ventricular ejection fraction <55%, or (5) left ventricular hypertrophy. In these cases, 6 hearts were successfully perfused with OCS and used for transplantation out of a total of 43 transplants performed over this time period, increasing the transplantation rate by 14%. In comparison to the non-marginal donors preserved with static cold storage during this time period, these extended-criteria donors with OCS preservation experienced similar hospital length of stay and overall survival. The results of these single center studies are encouraging, but the small sample size and non-randomized nature limit the conclusions that can be drawn.

### 3.4. XVIVO Non-Ischemic Heart Preservation

Early research in machine heart perfusion was centered around hypothermic oxygenated perfusion methods [14]. Wicomb and colleagues performed much pre-clinical work of hypothermic oxygenated perfusion, demonstrating successful preservation of pig and baboon hearts up to 48 h [68,69]. Additionally, they were the first to report a clinical application of hypothermic perfusion in human allotransplant as early as the 1980s [70], though with only one patient surviving past 16 months. 

In today’s practice, the XVIVO Heart Preservation System (XVIVO Inc, Gothenburg, Sweden) (XHPS) is an additional system currently under Phase II clinical trial (Figure 3). This system, instead of warm donor blood perfusion, utilizes a portable device in which the heart is perfused with a cold (8 °C), a nutrient- and hormone-enriched cardioplegic solution containing red blood cells. In preclinical study, this system was able to preserve porcine hearts for up to 24 h [71,72], and furthermore, improved outcomes when utilized in place of cold static preservation during pig-to-baboon xenotransplantation [73,74].

#### 3.4.1. Device Setup

Prior to cardiectomy, the XHPS reservoir is filled with 2.5 L of perfusion solution. Additionally, 500 mL of donor/recipient compatible irradiated and leukocyte-reduced blood is added (from hospital blood bank). This achieves a system hematocrit of ~15%. The donor cardiectomy is performed under standard conditions. The heart is then attached to the XHPS via an aortic cannula. Once mounted, the heart is then submerged into the reservoir of blood and perfusion solution. The XHPS contains an in-series roller pump, oxygenator, leukocyte filter, and cooler/heater. Oxygenated blood is pumped into the aortic root to maintain a pressure of 20 mmHg and coronary artery flow between 150–250 mL/min. The heater/cooler maintains a temperature of 8 °C, and the pH is maintained at 7.4 [75].

#### 3.4.2. Clinical Usage and Applications

Though less established within clinical heart transplantation, XVIVO technologies have shown much promise in the field of ex vivo lung preservation and evaluation of extended-criteria lung donors for transplantation [76]. In an initial trial utilizing ex vivo perfusion of extended-criteria donor lungs with XVIVO technologies, graft utilization was 55%. In these transplanted recipients, one-year survival was comparable to a cohort of patients that received standard-criteria donor lung grafts without ex vivo perfusion. Early clinical applications of the XVIVO heart perfusion system have also been promising for donor heart preservation. In a non-randomized trial of 42 patients, 6 donor grafts were perfused with XHPS, and 25 were assigned to standard cold storage (11 patients were excluded) [75]. At the 6-month posttransplant timepoint, there were no deaths with XHPS and 4 (16%) deaths with static cold storage. The XHPS-perfused hearts also experienced a 100% rate of freedom from major events (severe primary graft dysfunction, posttransplant extracorporeal membrane oxygenation, or ≥2R acute cellular rejection). Secondary outcomes such as rates of renal or pulmonary failure, as well as intensive care unit length of stay were comparable between groups. Although clinical usage of this device is early and relatively small, these results suggest that the results from pre-clinical animal studies may be applicable to humans.

## 4. Donation after Circulatory Death

The median age and risk profiles of heart recipients have changed over the last few decades. Likewise, the age and comorbidity burden of donors have also evolved [1]. Numerous factors may contribute to these findings including an aging population, as well as improved helmets, seatbelts, and vehicle safety. For example, head trauma as a means of donor cause of death have decreased within North America and Europe, and other causes such as stroke and/or anoxia have increased. Additionally, the average age of potential heart donors has increased, along with the complexity of medical comorbidity and risk factors [1]. As the demand for donor hearts continues to rise, transplanting centers must start to consider the use of donors outside the previously established standard acceptance criteria. This also holds true in pediatrics, where mechanical circulatory support options remain unideal and substantial waitlist mortality persists for neonates, infants, and small children [77].

Along with utilization of extended criteria donors, renewed interest in the DCD population may offer a means of increasing the available donor pool. Largely abandoned since the development of DBD legislation, DCD for heart donation may offer an alternative donor pool, often younger and with a lower degree of comorbidity burden [78]. However, criticisms of DCD harvest include the ischemic injury that the heart must incur to achieve cardiac arrest once life support is withdrawn. As experience continues to grow with DCD procurement, methods of minimizing warm and cold ischemia times will likely continue to improve.

At present, two contemporary approaches to DCD harvest have been described. The first, as described by Messer and colleagues [79], entails cardiac arrest after withdrawal of life support and establishment of normothermic regional perfusion (NRP). In this, the heart is re-perfused in situ by placing the donor on veno-arterial extracorporeal life support or cardiopulmonary bypass, but with selective clamping or ligation of the cephalic vessels to prevent brain re-perfusion. This method closely resembles the first transplantation performed by Christiaan Barnard, in which the donor heart was re-perfused on cardiopulmonary bypass after initial arrest [16]. With the heart reanimated, extracorporeal support is weaned, and the heart is then evaluated and if acceptable, it is arrested using conventional cold cardioplegic arrest. The heart is then either stored via standard cold storage or re-perfused with the Transmedics OCS system and transported back to the transplanting center. An advantage to this technique is the ability to assess the donor heart in situ after re-perfusion and reanimation of the heart. When using static cold storage techniques after the heart is removed, this technique obviates the need for expensive machine perfusion devices such as the Transmedics OCS system, which may allow DCD heart transplantation to be economically feasible for many transplanting centers. A potential downside of this method, however, are the conflicting viewpoints with regard to its ethicality [80]. One main criticism is that brain reperfusion may still occur despite ligation of cephalic vessels by way of spinal artery collaterals. At present, there remains to be a universal consensus with regard to the acceptability of NRP across countries and transplanting centers.

In the second method, first described by Dhital [22], and later utilized by Garcia Sáez [81], involves direct procurement of the donor heart after cardiac arrest. In this method, cardiac arrest is observed after withdrawal from life support. Herein, there is no opportunity to re-evaluate the function of the heart in situ after cardiac arrest occurs. The heart is then grossly accessed, and cold cardioplegia is delivered to the aortic root and coronary arteries without in situ reperfusion. The cardioectomy is performed, and the heart is then re-perfused ex vivo using the Transmedics OCS system. Evaluation of donor graft function is assessed ex situ prior to transplantation.

The DCD experience utilizing NRP has been promising. Experiences from the United Kingdom, Belgium, and Spain have suggested that NRP with static cold storage, without the need for ex situ perfusion, may reduce the overall cost of organ procurement, as well as the complexity of the procurement procedure [82,83,84]. In a report by Minambres and colleagues, four heart transplants were performed using NRP with static cold storage for transport. In all cases, the recipients were discharged home with excellent outcomes [83]. In the United States, early reports of DCD donation outcomes have been encouraging, including an 8-patient series described by Smith and colleagues using NRP, including one heart-lung transplant and one heart-kidney transplant [85]. Hoffman and colleagues also published a 15 patient series of DCD transplant with NRP [86]. In this series, thirty-day survival was 100%, and the rates of mild and moderate primary graft dysfunction were 40% and 20%, respectively.

The results of direct DCD procurement with ex situ reperfusion have also been promising. In the 3-patient series of DCD donation by Dhital [22], utilizing re-perfusion with the Transmedics OCS, 2 of the transplanted recipients required temporary mechanical circulatory support following transplantation. However, all 3 patients were doing well at one week. In a 2-patient series described by Garcia Sáez [81], both patients had previous indwelling left ventricular device. Postoperative length of stay for these patients were 62 days and 46 days, respectively, but the recipients had normal left ventricular function. At a follow up of 291 days and 290 days, both patients were alive and well. The largest series by Messer and colleagues [87], who published their 5-year experience of DCD donation. In this time, a total of 79 transplants were performed using DCD donation (22 NRP, 57 with direct procurement and OCS perfusion), increasing the center’s transplanting activity by 48% (164 DBD transplants performed). Thirty-day (DCD, 97% vs. DBD, 99%; *p* = 1.00) and one-year (DCD, 91% vs. DBD 89%, *p* = 0.72) survival were equivalent between cohorts. DCD donation with OCS re-perfusion has also been described in the pediatric population in a five patient series, in which all patients had excellent graft function on follow-up [88]. While preliminary, these results compare favorably to earlier pediatric DCD experience with cold storage, which reported significant early mortality [77]. The United States Food and Drug Administration has approved the usage of the Transmedics OCS system for heart preservation after DCD donation in April of 2022. Currently, a clinical trial utilizing the Transmedics OCS system after DCD donation has finished enrollment (clinicaltrials.govNCT03831048) [89].

These early results of DCD donation are promising and may be a possible avenue back to a largely untapped donor pool. Current estimates predict that widespread utilization of DCD donation may increase yearly adult heart transplants by 300–600 transplants/year in the United States alone [90,91]. As this has possibility has only recently begun to be re-explored, long-term performance of DCD grafts remains relatively unknown.

## 5. Ex Vivo Therapeutic Interventions

Ex vivo perfusion systems also may serve as an important vector to which therapeutic interventions can be administered to donor hearts without affecting the remainder of either donor or recipient organs. Such interventions aim to reduce the impacts of ischemia/reperfusion injury, but may also allow marginal organs that would otherwise be declined to be “re-conditioned” for acceptable usage [15]. Though these therapies have largely not been tested in a clinical arena, their re-clinical findings have shown promise. Eventual adoption of these ex vivo therapies may help expand the current pool of donor organ availability.

Oxygen-derived free radicals play a large role in ischemia/reperfusion injury [92], and myocyte loss is mediated largely through cell apoptosis, autophagy, and necrosis [93]. The apoptosis pathway has been long studied, with therapeutic interventions aimed at inhibiting its key enzymes. For example, Wei and colleagues have shown that blockade of this pathway through perfusion of pig hearts with hypothermic solution containing small interfering RNA molecules targeting expression of key apoptotic and inflammatory enzymes (complement, caspase-8, caspase-3 and nuclear factor κB-p65 genes) resulted in reductions in cell apoptosis and myocyte damage. Donor heart function was improved in this porcine heart transplant model [94]. Furthermore, the perfusion of pig hearts with oxygen-derived free-radical scavengers such as MCI-186 have also been shown to improve graft function and reduce cellular edema in another porcine transplant model [95]. While apoptosis was once considered the only regulated form of cell death, recent advances have identified that some forms of cell necrosis, or necroptosis, are at least in part regulated. This necroptosis is mediated through the mitochondrial response to oxidative stress associated with reperfusion injury [96]. A necroptosis pathway inhibitor, necrostatin-1, along with an analog, (Z)-5-(3,5-dimethoxybenzyl)-2-imine-1-methylimidazolin-4-1, have been both shown to reduce the effect of ischemia/reperfusion injury in rat cardiomyocytes [97,98]. Lastly, other therapies promoting angiogenesis, such as the addition of vascular endothelial growth factor [99], prokinectin receptor-1 [100], human multipotent stromal cells [101] have been shown to increase cardiomyocyte survival after ischemic insult.

In addition to therapies aimed at reducing ischemia/reperfusion injury, heart perfusion systems may also a unique approach to deliver donor heart-specific gene therapy. In a model presented by Bishawi and colleagues [101], an adenoviral vector was delivered to a porcine heart while perfused with the Transmedics OCS system. These hearts were then transplanted heterotopically into recipient pigs. After five days, vector gene expression levels were high throughout the transplanted allograft, but minimal among native recipient tissues. This technology may allow for biological modification of donor grafts prior to transplantation in the future.

## 6. Conclusions

Heart transplantation is the optimal treatment of end-stage heart failure for many patients, and the demand for donor organs is likely to continue to outgrow the available supply. Successful cardiac transplantation is dependent on the foundation of efficient organ procurement with adequate en route preservation. Heart preservation aims to reduce the harmful sequelae of myocardial ischemia to allow for safe transport and implantation. As current techniques and technologies continue to evolve, transplanting centers aim to push the boundaries on conventional practices and ischemic time restrictions. These new developments will not only allow for longer transport distances and a wider organ sharing network, but also expand the current donor pool through ex vivo evaluation of extended-criteria and DCD donor organs. Additionally, they may offer a means for donor organ-targeted therapies prior to transplantation.

## Figures and Tables

**Figure 1 jcm-11-05762-f001:**
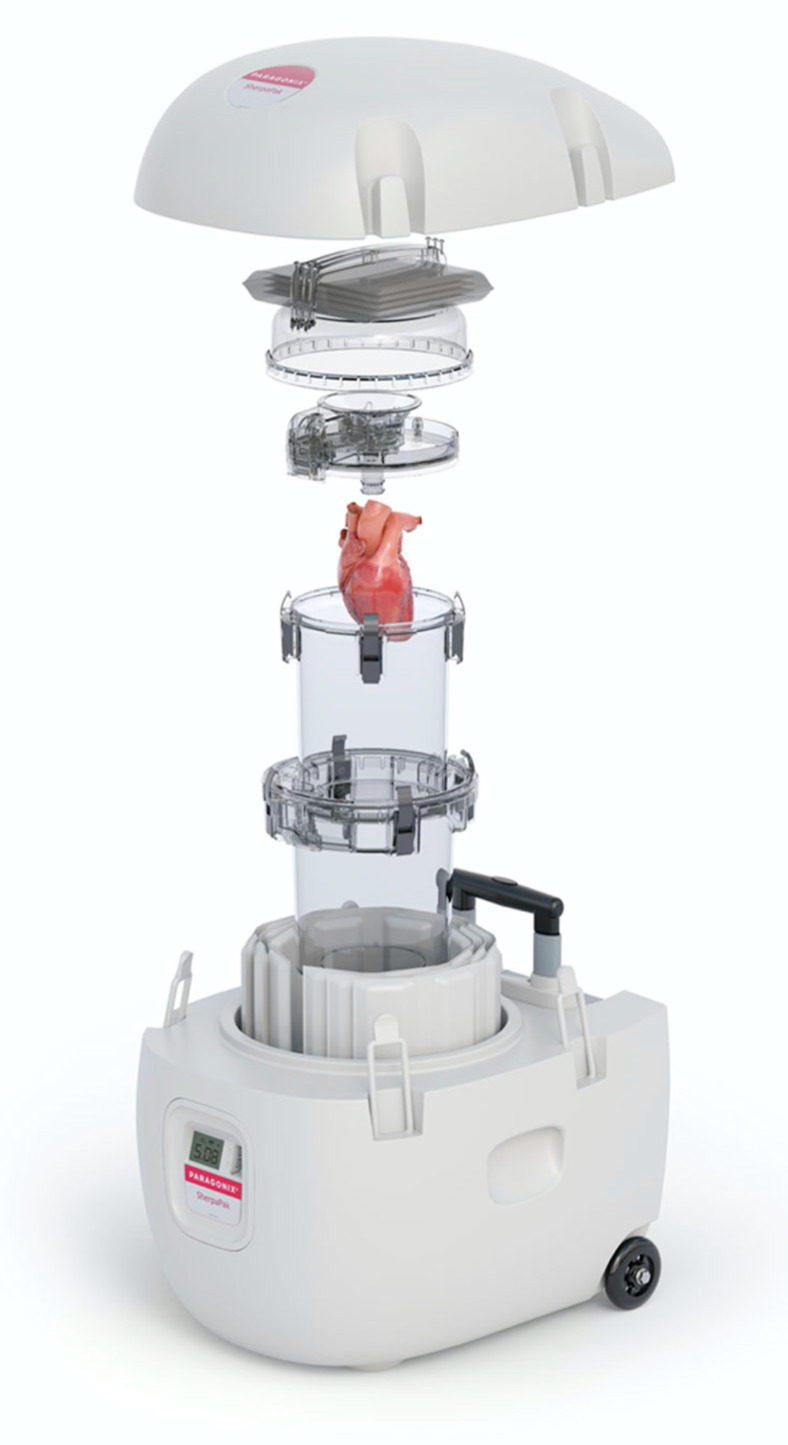
The Paragonix SherpaPak Cardiac Transport System. The heart is submerged in cold cardioplegia solution. The transport system uses proprietary phase change cold packs to maintain temperatures 4–8 °C. Photo used with permission from Paragonix Technologies Inc.

**Figure 2 jcm-11-05762-f002:**
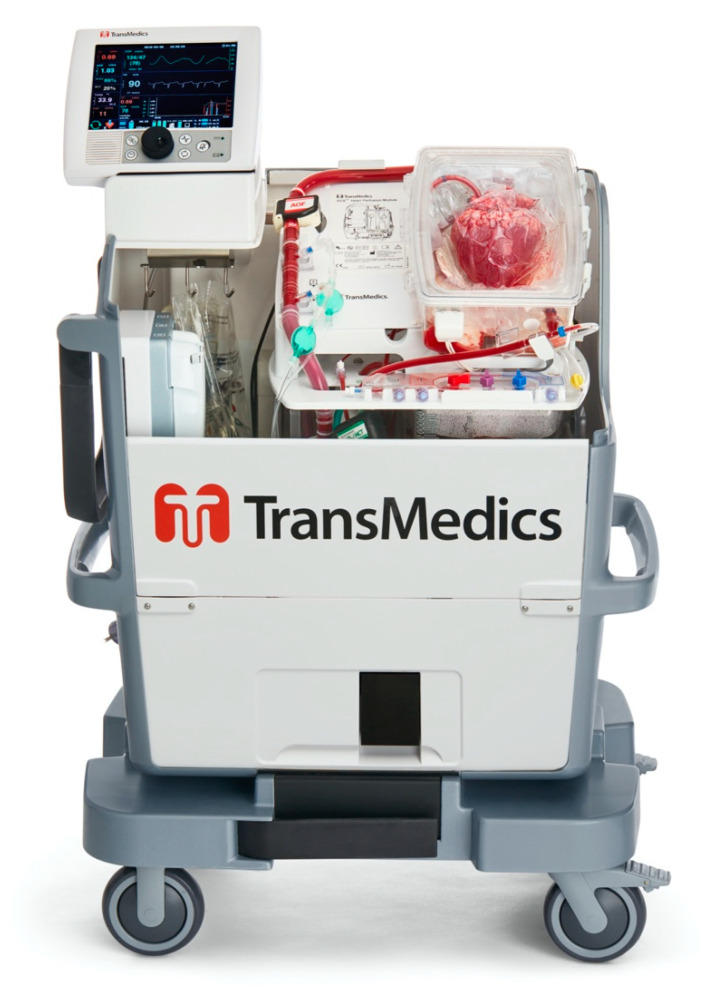
Transmedics Organ Care System. Normorthermic, continuous perfusion is established via the aortic root. The heart is maintained in sinus rhythm during transport. Photograph used with permission from Transmedics Inc.

**Figure 3 jcm-11-05762-f003:**
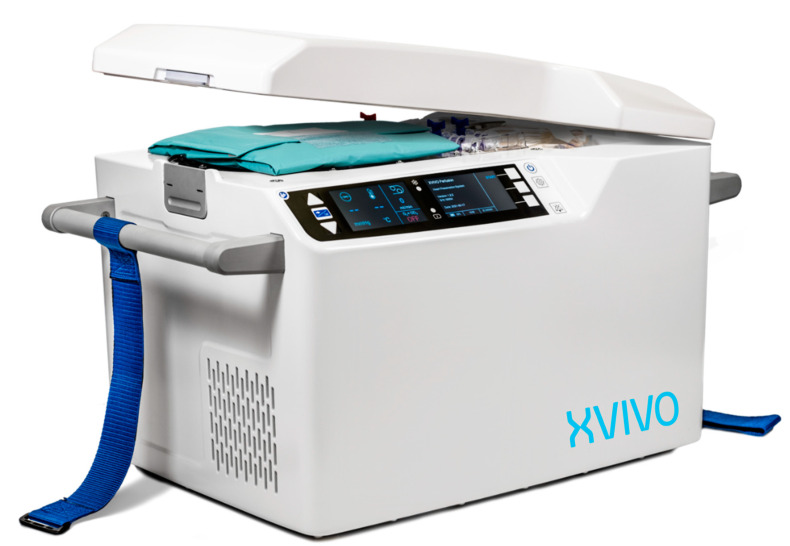
XVIVO Heart Perfusion System. The heart is submerged in cold blood and cardioplegia solution. Continuous, hypothermic perfusion is established via the aortic root during transport. Photograph used with permission from XVIVO Inc.

**Table 1 jcm-11-05762-t001:** Components and properties of commonly used heart preservation solutions.

Component (g/L)	UW	Celsior	HTK
Pentafraction	50	-	-
Lactobionic Acid	35.83	28.664	-
Potassium Phosphate monobasic	3.4	-	-
Magnesium Sulfate heptahydrate	1.23	-	-
Raffinose pentahydrate	17.83	-	-
Adenosine	1.34	-	-
Allopurinol	0.136	-	-
Glutathione	0.922	0.921	-
Potassium Hydroxide	5.61	-	
Mannitol	-	10.930	5.4651
Glutamic Acid	-	2.942	-
Sodium Hydroxide	Adjust to pH 7.4	4.000	-
Calcium chloride dihydrate	-	0.037	0.0022
Potassium chloride	-	1.118	0.6710
Magnesium chloride hexahydrate	-	2.642	0.8132
Histidine	-	4.650	27.9289
Histidine monohydrochloride monohydrate	-	-	3.7733
Hydrochloric acid	Adjust to pH 7.4	-	-
Sodium chloride	-	-	0.8766
Potassium hydrogen 2-ketoglutarate	-	-	0.1842
Tryptophan	-	-	0.4085
**Physical properties**			
pH	7.4	7.3	7.2
Osmolarity (mosmol/kg)	320	320	310

g = gram; HTK = histidine-tryptophan-ketoglutarate; kg = kilogram; L = liter; mosmol = milliosmole; pH = potential of hydrogen; UW = University of Wisconsin.

## Data Availability

No new data were created or analyzed in this study. Data sharing is not applicable to this article.

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
