# Peer review of "Heart Donation and Preservation: Historical Perspectives, Current Technologies, and Future Directions"

_jcm, 2022, doi:10.3390/jcm11195762_

Round 1

Reviewer 1 Report

This is an interesting and well-read review of the field. It is very much USA- orientated, although does describe work done in Sweden, Australia and the UK. But the focus is on the implications of changing allocation rulings in the US. There is nothing, apart from the DCD section, about how organ perfusion might allow an enlarged donor pool, perhaps with the use of more marginal, and particularly older donors

However, it comes after two recent reviews of much the same field. There is no reference to the paper of Wang et al, Ex-situ Heart Perfusion – The Past Present and Future. J Heart Lung Transplant 2021;40:69−86, or Qin et al, Machine Perfusion for Human Heart Preservation: A Systematic Review, Transplant International, doi;10.3389/ti 2022 10258.

Some of the history is a bit superficial; the hypothermic perfusion work of Wicomb and the auto-perfusing study of Hardest, both clinical applications, should be mentioned. It is good to see the Sherpa-Pak device described. But the inclusion. of the Paragonix Sherpa Perfusion, which has only been used in pigs, is a bit of a mystery. There are other devices which have reached the same large animal experimental stage

There is good background to DCD heart retrieval - Dhital preceded Garcia-Saez, and this should be corrected. The pros and cons of TA_NRP need expansion, perhaps with reference to the work done in Belgium and Spain

But there are a number of important omissions. Paediatric retrieval and preservation is not really touched upon, nor is the scope of therapeutic interventions 

Author Response

We thank you for these detailed and thoughtful requests which we feel will greatly enhance the quality of our manuscript. We will address your points in chronological order below:

  1. We acknowledge the oversight that our manuscript was overly focused on work and outcomes in the United States, and as a result may not have been as applicable and well-received as it otherwise could be by an international readership. We have revised the manuscript in several locations to highlight the global nature of our field and some of the very important work done in other countries. In particular, we have modified the abstract and introduction to reflect global heart transplant outcomes, and added a number of additional citations highlighting work from Belgium, Spain, Japan, China, and others.
  2. We also acknowledge that, while an important topic, there was perhaps too much sole focus on the implication of these technologies on the new organ allocation system in the United States for a global readership. As requested, we have added additional text and reference regarding how these technologies may enable expansion of the donor pool.
  3. We thank the Reviewer for bringing to our attention Wang et al and Qin et al – these represent two excellent recent reviews in the field. We have added references to these papers as requested.
  4. We thank the Reviewer for the suggestion to add more early historical work in the field, in particular that of Wicomb and Hardesty. We have added additional text and citations to this work as requested.
  5. We thank the Reviewer for pointing out the inconsistency of us including the Paragonix Sherpa Perfusion device, which is early stage and has only been used in pre-clinical animal models. We have accordingly removed discussion of this device from the text.
  6. We apologize to the Reviewer for our oversight in inferring that Dhital was after Garcia-Saez. We have corrected the text accordingly to make it clear that Dhital’s landmark Lancet paper was the first to describe this technique.
  7. We thank the Reviewer for the request to expand the “pros and cons” of TA_NRP. We have added additional text to do this, while simultaneously adding references to the work done in Belgium and Spain in this regard.
  8. We thank the Reviewer for pointing out that the manuscript could benefit from more discussion of pediatric heart retrieval and preservation. In addition to our previous inclusion of early OCS preservation experience in pediatrics and pediatric DCD donation with use of OCS re-perfusion, we have added several additional discussions of pediatric heart transplant donation and preservation.
  9. We acknowledge that a description of the scope of therapeutic interventions that can be employed syntropically with these technologies was not discussed, which we agree would greatly enhance the manuscript. We have added a new section to the manuscript to address this.

Reviewer 2 Report

A comprehensive well written review of heart donation and preservation.  Advances in the field have reignited DCD donation and more centers are utilizing technology to perform a larger number of transplants.   Minor suggestion:  

1. Section 3.4.2  Clinical Usage and Applications

Paragraph makes a rapid transition from discussing lung preservation in one sentence (ref 70) to discussing heart transplants that is not clear.  In reading the next sentence, "In a non-randomized trial of 42 patients, 6 donor grafts..." I initially thought you were still discussing lung transplant and had to look up the reference.  Please provide a smoother transition and clarification for the reader. 

Author Response

We thank you for your excellent suggestion that we feel will enhance the quality of our manuscript. We agree that the wording of this section was unclear and potentially confusing to the reader. We have modified this text to provide a smoother transition and clarification for the reader as requested.

Round 2

Reviewer 1 Report

This is a much improved paper. This review is delighted that so many of the suggestions have been adopted

There remain some other comments which could lead to additional changes

Whilst in their notes, the authors appreciate that there have been two recent reviews on the same topic. The paper of Wang et al is mentioned only in the section on reconditioning and the other paper could not be seen – the track changes did not extend to the bibliography. I think the papers should be mentioned in the introductory section, to highlight the significant interest there now is in this field

Again, in the introduction, they might explain that optimal donor-recipient pairing is limited by ischaemic time

It is a useful reminder to include the description of Barnard’s first transplant. But the relevance can be heightened by pointing out that placement of the donor on bypass to resuscitate the donor heart was a direct precursor of TA-NRP from 50 years previously!

There is brief mention of expansion of the donor pool. Perhaps this could be illustrated by an additional brief section describing the link of ischaemia time and age. Older donors are less tolerant of longer ischaemic times – there is good evidence for this. If EVHP can reduce or almost eliminate the ischaemic time, use of older donors, of whom, particularly in Europe, there are large numbers, could make a substantial difference to the donor pool size.

Author Response

Once again, we thank you for the opportunity to revise our current manuscript. We have incorporated all of the helpful comments and suggestions, which have helped to strengthen the message of this manuscript. The following changes have been made to this manuscript.

  1. We have cited both review papers by Wang et al and Qin et al within the Introduction to highlight the interest in this field of work.
  2. We have added additional language to the introduction to better clarify that optimal donor-recipient matching is largely limited by ischemia time within heart transplantation.
  3. We have made references to Barnard’s first transplant and its similarity to the existing NRP protocol within the section describing NRP.
  4. In the Introduction, we have further extrapolated on the notion that farther donor-recipient distances are requiring longer ischemia times. This is also in the context that overall median donor age is increasing over time, especially among European countries (with reference to the 2020 Internal Thoracic Organ Transplant Registry of the ISHLT 37th adult heart transplant report). Further we have discussed that problem is that older donor organs are more negatively impacted by long ischemia times (with reference to work by John et al and Russo et al), highlighting the need for improved heart preservation and perfusion techniques to address this problem.